# Biochemical neuroplasticity in the cerebellum after physical exercise: Systematic review and meta-analysis

Marcio Gonçalves Corrêa[1,☯], Thais Alves Lobão[1,☯], Gabriel Mesquita da Conceição Bahia [1], Erica Miranda Sanches Aires[1], Rebeca da Costa Gomes[1*], Jefferson Hildo Medeiros de Queiroz [1], Marta Chagas Monteiro[2], Carlomagno Pacheco Bahia [1*]

**1** Laboratory of Neuroplasticity – Health Institute Sciences, UFPA, Pará, Brazil, **2** Laboratory of Laboratory Immunology, Microbiology, and In Vitro Assays – Health Institute Sciences, UFPA, Pará, Brazil

☯ These authors equally contributed to this work.
* carlomagno@ufpa.br, carlomagnobahia@gmail.com (CPB); rebecadcgomes@gmail.com (RDCG)

## Abstract

### Background

Neuroplasticity is the central nervous system's (CNS) capacity to adapt to injuries or environmental changes. Biochemical neuroplasticity is one such adaptation that may occur in response to physical exercise (PE). This systematic review and meta-analysis aimed to evaluate the effects of PE on cerebellar biochemical neuroplasticity.

### Methods

Following the PICO strategy, this review included in vivo studies with small rodents (Population) subjected to well-defined PE protocols (Intervention) and compared to non-exercised controls (Comparator) to assess cerebellar biochemical alterations (Outcome). Studies published between January 1976 and July 2024 without language restrictions were searched in PubMed, Scopus, Web of Science, and Cochrane Central databases. Data were synthesized through meta-analyses and methodological quality was assessed by the SYRCLE risk of bias tool.

### Results

Out of 3,107 records screened, six studies met the inclusion criteria for qualitative and quantitative analyses. All studies had a low or unclear risk of bias. Markers of biochemical neuroplasticity assessed included superoxide dismutase (SOD), catalase (CAT), glutathione (GR), reduced glutathione (GSH), glutathione peroxidase (GSH-Px), glutathione disulphide (GSSG) and lipid peroxidation (LPO). Meta-analyses showed that moderate-volume PE significantly reduced LPO (SMD = −2.41; 95% CI: −3.89 to −0.93), while high-volume PE increased LPO (SMD = 4.55; 95% CI: 1.92 to 7.18). Low-intensity or low-volume PE did not significantly alter oxidative markers.

**Data availability statement:** All relevant data are within the manuscript and its Supporting Information files.

**Funding:** CPB: National Council for Scientific and Technological Development – CNPq (grants no. 310054/2018-4, 447835/2014-9, 483404/2013-6, 444967/2020-6, and 444982/2020-5), and the Brazilian Agency for Support and Evaluation of Graduate Education – CAPES (grants PROCAD 21/2018). MCM: National Institutes of Science, Technology and Innovation (PROBIAM Pharmaceuticals Amazonia - INCT/CNPq grant 406819/2022-0), the Brazilian Agency for Support and Evaluation of Graduate Education – CAPES (88882.461690/2019-01), and the Fundação Amazônia Paraense de Amparo à Pesquisa (FAPESPA) grant 005/2016.The funders had no role in study design, data collection and analysis, decision to publish, or preparation of the manuscript.

**Competing interests:** The authors have declared that no competing interests exist.

**Abbreviation:** EtOH, ethanol; PE, physical exercise; PEg, physical exercise group; CTL, control group; Mv.PE, moderate-volume physical exercise group; Hv.PE, high-volume physical exercise group; CNS, central nervous system; SYRCLE, Systematic Review Center for Laboratory Animal Experimentation; SOD, superoxide dismutase; CAT, catalase; GR, glutathione; GSH, reduced glutathione; GSH-Px, glutathione peroxidase; GSSG, glutathione disulfide; LPO, lipid peroxidation

## Conclusions

PE induces either adaptive or maladaptive biochemical neuroplasticity in the cerebellum depending on protocol variables. While enzymatic activity responds to cellular changes and limits nervous tissue protection, adaptive biochemical neuroplasticity seems to confer greater resistance and efficiency.

## Introduction

Neuroplasticity refers to the CNS's ability to reorganize and adapt structurally and functionally in response to environmental stimuli or injury [1,2]. At the cellular and molecular levels, neuroplasticity encompasses alterations in neuronal structure and function, including synaptic remodeling and pathways reorganization [1]. These changes can be beneficial (adaptive neuroplasticity) or detrimental (maladaptive neuroplasticity) depending on their impact on CNS integrity and function [3,4]. Biochemical neuroplasticity refers to neurochemical alterations in response to stimuli such as physical exercise (PE), which can lead to oxidative stress that affects neuronal integrity [5]. Although neuroplasticity encompasses several neuronal events, the terms adaptive or maladaptive biochemical neuroplasticity can address changes in cerebellar oxidative biochemistry.

Adaptive biochemical neuroplasticity is associated with enhanced neuroprotection, memory, and motor function. Conversely, maladaptive responses involve an oxidative imbalance in the nervous tissue that damages the mitochondrial energetic function and proteins/enzymes and lipids of the cell membrane, which leads to oxidative stress, cell death, and functional damage [6]. Environmental stimuli [7], food [8], alcohol consumption [9], and PE can induce biochemical neuroplasticity [10,11].

PE-associated improvements in cognitive domains such as memory [12], reaction time [13], and motor learning [14] may result from the modulation of enzymes, including superoxide dismutase (SOD), catalase (CAT), glutathione reductase (GR), glutathione disulfide (GSSG), and glutathione peroxidase (GSH-Px), as well as changes in lipid peroxidation (LPO) across various brain regions, such as the motor cortex, hippocampus, brainstem, and cerebellum [15,16]. However, these health benefits are dependent on the intensity, volume, and frequency of PE [17,18].

PE is strongly associated with cerebellar functions that regulate postural control, coordination, planning, learning, and motor execution [19]. The high energetic demand of cerebellar neurons – required to receive and project axons to various cortical areas including posterior parietal, prefrontal, and primary motor regions – increases the formation of reactive oxygen species (ROS). Consequently, the cerebellum is particularly susceptible to biochemical alterations resulting from the increased energy expenditure induced by PE [20].

Although low-intensity PE surprisingly does not appear to induce cerebellar neuroplasticity [21], high volumes of moderate-intensity PE may result in maladaptive biochemical responses and neural tissue damage [10]. Thus, professionals employing PE for high-performance sports training, health promotion, or therapeutic

purposes must have a clear understanding of the relationship between PE parameters and biochemical neuroplasticity. This systematic review and meta-analysis were therefore conducted to evaluate the effects of PE on cerebellar biochemical neuroplasticity.

## Materials and methods

This systematic review and meta-analysis was registered in the Open Science Framework (OSF) database under record 10.17605/OSF.IO/ERBD2 (access at https://osf.io/erbd2/) and followed the Preferred Reporting Items for Systematic Reviews and Meta-Analyses (PRISMA 2020) guidelines (see S1 File) [22].

### Inclusion and exclusion criteria

This review followed the PICO strategy:

**Population (P).** Experimental studies involving small rodents

**Intervention (I).** PE protocols with well-defined volume, intensity, and frequency.

**Comparator (C).** Small rodents not subjected to PE.

**Outcome (O).** Biochemical changes in cerebellar nervous tissue following PE.

**Study design.** Eligible studies were original in vivo experimental investigations that assessed the biochemical effects of PE on the cerebellum of small rodents using well-defined training protocols in terms of volume, intensity, and frequency.

**Inclusion criteria.**

(1) Title and/or abstract containing relevant descriptors.

(2) No language restrictions.

(3) Original studies.

(4) In vivo models with small rodents.

(5) Evaluation of cerebellar biochemical changes resulting from PE protocols with well-defined volume, intensity, and frequency.

**Exclusion criteria.** Reviews, case reports, descriptive studies, opinions, technical articles, guidelines, and in vitro studies were excluded.

### Search strategy

The MeSH descriptors "animal", "brain", "exercise", "physical fitness", and "oxidative stresses" were used. The search was slightly adapted for each database as detailed in S2 and S3 Files. Search alerts were configured to notify authors of newly published studies. Two reviewers (MGC and TAL) conducted independent database searches.

### Information sources

Studies published between January 1976 and July 2024 were searched in PubMed, Scopus, Web of Science, and Cochrane Central, with no language restrictions.

### Screening and data extraction

Duplicate records and those lacking relevant MesH descriptors in the title or abstract were excluded using the Rayyan platform (https://new.rayyan.ai/). Full-text reading and selection were independently performed by two reviewers (MGC

and TAL), with disagreements resolved by a third reviewer (CB) (S2 and S3 Files). Extracted data included: author, study, design, exercise protocol, enzymatic assessment, oxidative damage assessment, method of stress induction, results, conclusions, reviewer, date, and third reviewer (S4 File). A summary of the included studies is provided in Table 1.

### Critical appraisal of individual sources of evidence

Methodological quality and risk of bias were independently evaluated by two reviewers (MGC and TAL) using the SYR-CLE tool, which is adapted from the Cochrane RoB tool for animal studies (S5 File) [23]. Ten domains were assessed: (a) sequence allocation; (b) baseline group similarity or confounders adjustment; (c) random allocation; (d) random housing conditions; (e) blinding of caregivers/investigators; (f) random outcome assessment, (g) blinding of outcome assessor; (h) handling of incomplete data; (i) absence of selective outcome reporting; and (j) other sources of bias. Each domain was evaluated by answering several questions namely 'yes', 'probably yes', 'no', 'probably no', 'no information', or 'not applicable', and then rated as 'high', 'low', or 'unclear'. Studies were deemed low risk of bias if all domains were rated as low, while studies rated as high or unclear for at least one domain were deemed high or unclear risk of bias, respectively. This tool assessed whether the methods were adequate to provide consistent and valid information, as well as whether the results revealed the expected effects.

### Data analysis

**Descriptive synthesis.** We summarized findings for SOD, CAT, GSH, and LPO based on structured extraction from original studies investigating biochemical changes in cerebellar tissue after PE.

**Quantitative synthesis.** When two or more studies were available, meta-analyses were conducted for SOD, CAT, GSH, and LPO by using review manager software (RevMan 5.4, The Cochrane Collaboration; Copenhagen, Denmark) [24]. Meta-analyses were stratified by PE volumes and only considered the studies in which the animals completed the PE protocols. Sample size, means, and standard deviations were used to compare groups (PE versus control). When raw data were only available in graphical form (i.e., data also absent in supplementary material), numerical estimates were derived using validated software (WebPlotDigitizer) [25]. Effect sizes were computed using a random-effects model and the inverse variance method. The standardized mean difference (SMD) and 95% confidence intervals (CIs) (p < 0.05) were reported. Effect sizes were interpreted as small (SMD < 0.40), moderate (SMD = 0.41–0.70), or large (SMD > 0.70) (26–28). Heterogeneity was assessed using the $I^2$ statistic and interpreted as low (~25%), moderate (~50%), or high (~75%) [24]. Funnel plots were used to assess potential publication bias (see S6 File), and forest plots displayed effect sizes for each outcome [24–28].

## Results

### Study selection

From 3,107 initial records, 767 duplicates were excluded and 2,330 records were excluded after title and abstract screening. Among 10 records selected for full-text reading, one study was excluded for lacking an exercise group, one study was excluded due to the absence of cerebellar oxidative stress-related outcomes, and two studies were excluded for evaluating spontaneous PE. Finally, six animal studies were included for qualitative and quantitative analysis [10,14,15,21,29,30], see Fig 1 and S3 File.

### Study characteristics

SOD was evaluated in five studies [10,15,21,29,30], CAT was assessed in four studies [10,15,29,30], GSH was determined in four studies [10,14,15,30], GSH-Px was evaluated in two studies [15,30], GSSG was assessed in three studies [10,15,30], and GR was determined in two studies [15,30]. The six studies evaluated LPO [10,14,15,21,29,30] and two studies used ethanol to induce oxidative stress [14,15].

**Table 1. Data extracted from the selected articles.**

| Author | Design | PE protocol | Enzyme assessment | Oxidative damage assessment | Stress induction | Results | Conclusions |
|---|---|---|---|---|---|---|---|
| Somani et al., 1997 | Adult male Fischer-344 rats: CTL (n = 6) PEg (n = 6) PE + EtOH (n = 6) | - Treadmill inclination: 6° - Duration: 6 weeks - Frequency: 5 days/week - Volume: 5–30 min/day - Intensity: 8–19 m/min | SOD CAT GSH GSH-Px GSSG GR | LPO | EtOH 20% | ↓SOD (PEg and PE + EtOH) θ CAT θ GSH/GSSG ↓GSH-Px (PEg) θ GR (PEg) ↑GR (PE + EtOH) ↓LPO (PEg and PE + EtOH) | PE modulated oxidative bio-chemistry and reduced EtOH-induced oxidative stress, albeit did not changed control animals. |
| Chalimoniuk et al., 2015 | 5 to 6-week-old male Wistar rats: CTL (n = 18) PEg (n = 18) | - Treadmill inclination: 0° - Duration: 6 weeks - Frequency: 5 days/week - Volume: 40–60 min/day - Intensity: 16 m/min (1° week), 20 m/min (2° week), 24 m/min (3° week), 28 m/min (4°, 5°, and 6° weeks) | SOD CAT GSH GSH-Px GSSG GR | LPO | – | θ SOD ↑ CAT ↑ GSH θ GSH-Px θ GR ↑ LPO | PE modulated oxidative bio-chemistry, but did not prevent PE-related oxidative stress. |
| Casuso et al., 2015 | 6-week-old male Wistar rats: CTL (n = 8) PE (n = 8) | - Treadmill inclination: 10° - Duration: 6 weeks - Frequency: 5 days/week - Intensity: Speed (m/min) + Time (min) 1° day: 20 min 2° day: 20 min 3° day: 25 min 1°, 2°, 3°, 4° week: 5 day/week: 80 min 6° week: 80 min | SOD CAT | LPO | – | ↓ SOD ↓ CAT ↓LPO | PE modulated oxidative bio-chemistry and reduces oxidative stress. |
| Lamarão-Vieira et al., 2019 | 30-day-old male Wistar rats: CTL (n = 10) PEg (n = 10) CTL + EtOH (n = 10) PE + EtOH (n = 10) | - Treadmill inclination: 0° - Duration: 4 weeks - Frequency: 5 days/week - Volume: 30 min/day - Intensity: gradual increase from 2 to 8 m/min (1st week), 5–10 m/min (2nd week), 8–12 m/min (3rd week), and 10–15 m/min (4th week). | GSH | LPO | EtOH 20% | θGSH ↑LPO (CTL + EtOH) ↓LPO (PE + EtOH) θ LPO (PE) | PE reduced EtOH-induced oxidative stress, albeit did not changed control animals. |
| De Souza et al., 2020 | Adult 60-day-old male Wistar rats: CTL (n = 8) Mv.PE (n = 8) Hv.PE (n = 8) | - Treadmill inclination: 10° - Duration: 12 weeks - Volume: gradual increase from 10 to 30 min (moderate) or 10–90 min (high) - Frequency: 3–5 times/week - Intensity: 50–70% of maximum speed previously tested | SOD CAT GSH GSSG | LPO | – | ↑ SOD (Hv.PE and Mv.PE) ↑ CAT (Mv.PE) θ GSH θ GSSG ↑ LPO (Hv.PE) | High- and moderate-volume PE modulated oxidative bio-chemistry, but did not prevent PE-related oxidative stress. |
| Silveira et al., 2020 | Male Wistar rats 6 months/ Adult (n = 12) 18 months/ middle-aged (n = 12) | - Treadmill inclination: 0° - Duration: 12 weeks - Volume: 30 min/day - Frequency: 3 days/week - Intensity: gradual increase from 2 to 8 m/min (low) | SOD | LPO | – | θ SOD θLPO | PE did not modulate oxidative bio-chemistry and did not promoted changes in oxida-tive stress. |

↑: upregulation; ↓: downregulation; θ: no difference; EtOH: ethanol; PEg: physical exercise group; CTL: control group; Mv.PE: moderate-volume physical exercise group; Hv.PE: high-volume physical exercise group

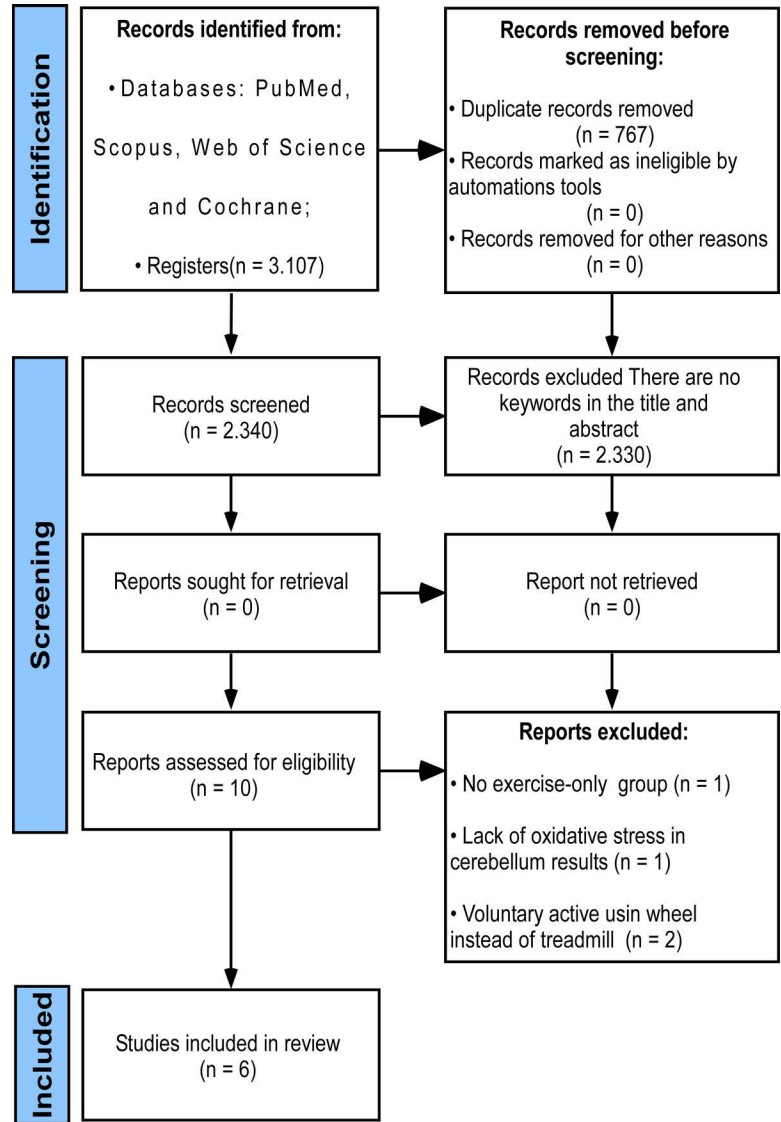

**Fig 1. Flowchart of the database search following the PRISMA statement.**

## Risk of Bias

The studies exhibited a low and unclear risk of bias for eight (a, b, c, d, f, h, i, and j) and two domains (e and g), respectively. No study was deemed at high risk of bias in any domain.

## Individual study results

In two studies, PE promoted cerebellar neuroprotection against ethanol-induced damage [14,15]. Low- and moderate-intensity PE were performed on running treadmills for four [14], six [15,29,30], or twelve weeks [10,21]. High-intensity PE was not evaluated in any study. The PE volumes ranged from 15 to 90 minutes per day, with a frequency of 3–5 days per week (Table 1).

Overall, PE increased the activity of SOD, CAT, GR, and GSH [10,15,30], while GSSG activity did not changed [9,13,14]. Furthermore, PE reduced LPO in four studies [10,14,15,30], increased LPO in two studies [10,30], and did not change LPO in one study [21] (Table 2).

## Quantitative analysis

The meta-analyses for high- and moderate-volume PE are respectively shown in Figs 4 and 5. For high-volume PE, both SOD and CAT activities were not significantly different from their respective controls, and high heterogeneity among studies was observed (SOD overall effect = 0.65; 95%CI 0.74, −1.48 to 2.97; $I^2$ = 87%/ CAT overall effect = 0.74; 95%CI 1.33, −2.18 to 4.85; $I^2$ = 93%). In addition, LPO was significantly higher than control, and moderate heterogeneity among studies was observed (LPO overall effect = 3.39; 95%CI 4.55, 1.92 to 7.18; $I^2$ = 43%) (Fig 4).

For moderate-volume PE, SOD, CAT, and GSH activities were not significantly different from their respective controls (SOD overall effect = 1.17; 95%CI −0.86, −2.29 to 0.58; CAT overall effect = 0.62; 95%CI −0.19, −0.79 to 0.41; GSH overall effect = 1.52; 95%CI 0.53, −0.15 to 1.22). The heterogeneity among studies was low for CAT and GSH ($I^2$ = 0% and $I^2$ = 20%, respectively) and high for SOD ($I^2$ = 84%). In addition, LPO was significantly lower than the control (LPO overall effect = 2.67; 95%CI −1.94, −3.37 to −0.51) and high heterogeneity among studies was observed ($I^2$ = 84%) (Fig 5).

## Discussion

This review applied eligibility criteria for study selection (Fig 1) methods to reduce the risk of bias (Figs 2 and 3), and quantitative syntheses (Figs 4 and 5) to provide valuable conclusions regarding PE-induced cerebellar biochemical neuroplasticity [22]. Experimental studies using animal models provide consistent and reliable evidence of the biochemical alterations underlying neuroplasticity and the effects of PE protocols on health conditions.

Neuroplasticity is typically preceded by biochemical alterations, which appear to be key mechanisms that can either improve neural function or cause cell death [6]. Alterations in enzymatic activities and consequential neurophysiological effects must be understood as biochemical neuroplasticity, the induction of which by PE was demonstrated in this review. Interestingly, variations in the PE protocol can either increase or decrease the activity of antioxidant enzymes and LPO [10]. Given the complexity of biochemical systems, PE can promote either adaptive or maladaptive biochemical neuroplasticity (Figs 4–6). As shown in the meta-analysis of moderate-volume PE (Fig 5), increased $O_2$ consumption for short periods can significantly increase ROS formation and prevent the reestablishment of neural homeostasis by enzymes. Nevertheless, moderate ROS formation also acts as signaling stimuli

**Table 2. Up- and downregulation of enzyme activity and oxidative stress markers.**

| Study | SOD | CAT | GSH | GSH-Px | GSSG | GR | LPO | Stress induction |
|---|---|---|---|---|---|---|---|---|
| Somani et al., 1997 | ↓ (PEg)<br>↓ (PE + EtOH) | θ | θ | ↓(PE) | θ | θ (PEg)<br>↑(PE + EtOH) | ↓(PEg)<br>↓(PE + EtOH) | EtOH |
| Chalimoniuket al., 2015 | θ | ↑ | ↑ | θ | − | θ | ↑ | − |
| Casuso et al., 2015 | ↓ | ↓ | − | − | − | − | ↓ | − |
| Lamarão-Vieira et al., 2019 | − | − | θ | − | − | − | ↑(CTL + EtOH)<br>θ(PEg)<br>↓(PE + EtOH) | EtOH |
| De Souza et al., 2020 | ↑(Mv.PE)<br>↑(Hv.PE) | ↑(Mv.PE) | θ | − | θ | − | ↑(Hv.PE)<br>θ(Mv.PE) | − |
| Silveira et al., 2020 | θ | − | − | − | − | − | θ | − |

↑: upregulation;↓: downregulation; θ: no difference; EtOH: ethanol; PEg: physical exercise group; CTL: control group; Mv.PE: moderate-volume physical exercise group; Hv.PE: high-volume physical exercise group.

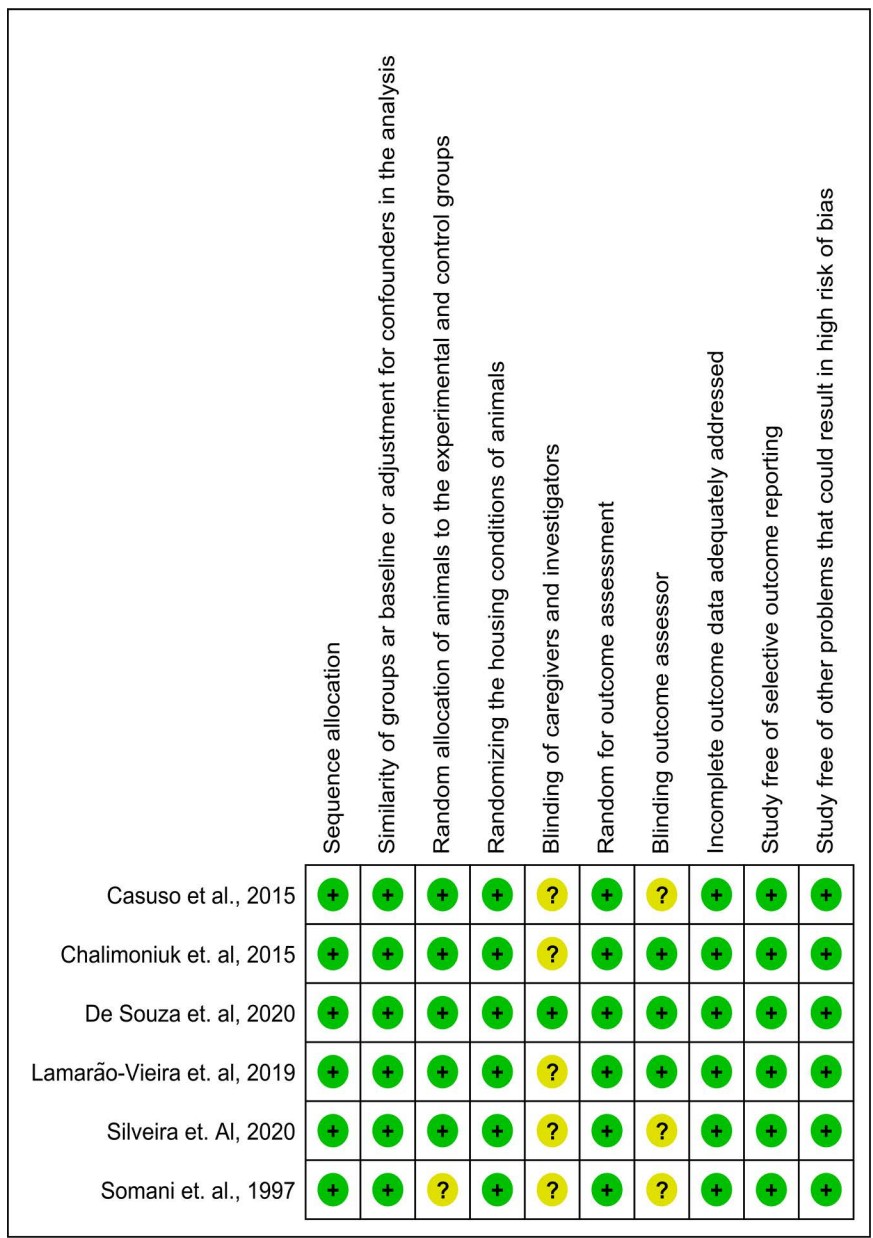

**Fig 2. Qualitative risk of bias assessment for included studies.** Green, yellow, and red circles indicate low, unclear, and high risk of bias, respectively.

that enhance antioxidant defenses [10,15,30] and affects adaptive neuroplasticity-related factors such as BDNF, FGF, VGF, and angiogenesis [31]. In contrast, excessive ROS formation due to prolonged $O_2$ consumption during high-volume PE (Fig 5) may overwhelm antioxidant systems, increase LPO, and cause oxidative stress, which impairs energy metabolism, induces cell death, or triggers neuroinflammatory responses – hallmarks of maladaptive biochemical neuroplasticity [9].

Modulation of LPO [10,14,15,29,30] and enzymes activities involved in cerebellar metabolic pathways – such as SOD, CAT, GR, GSH, and GSH-Px [10,15,29,30] – were reported in the selected studies as evidence of adaptive biochemical

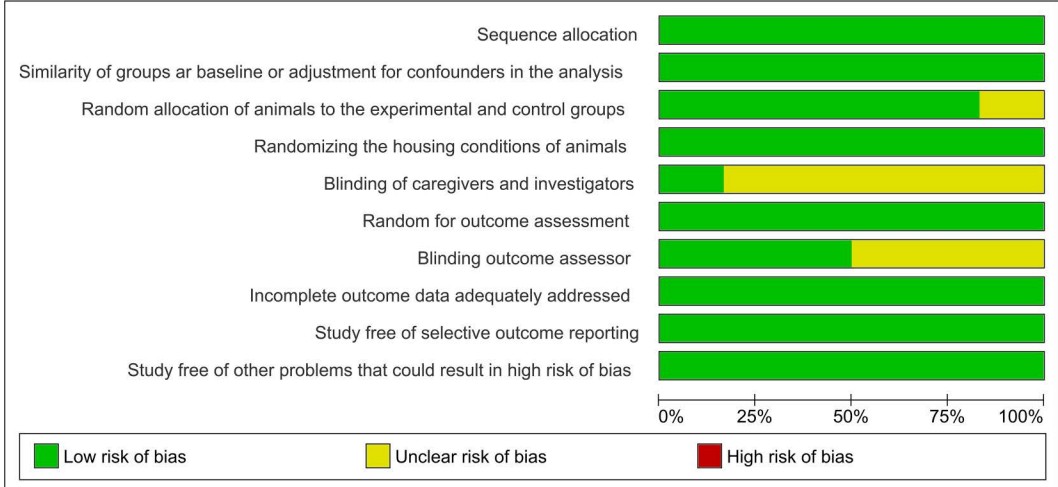

**Fig 3. Summary of reviewer's judgments for each domain presented as percentages.**

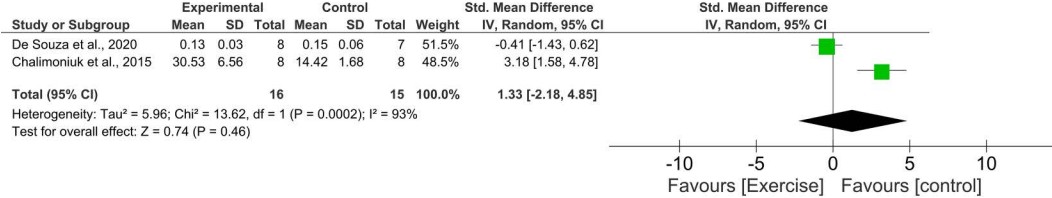

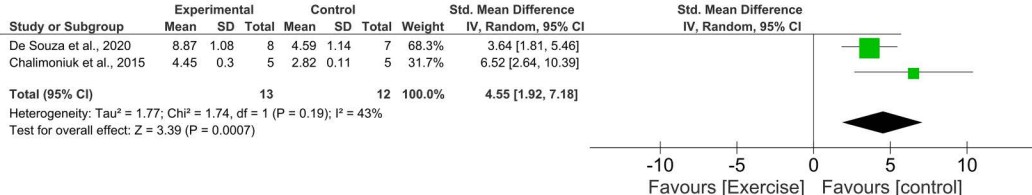

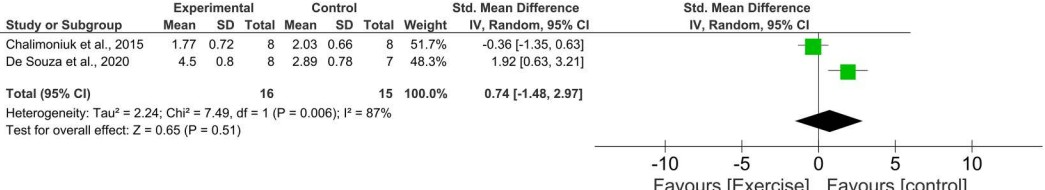

**Fig 4. Meta-analysis of high-volume PE groups versus controls.**

# SOD

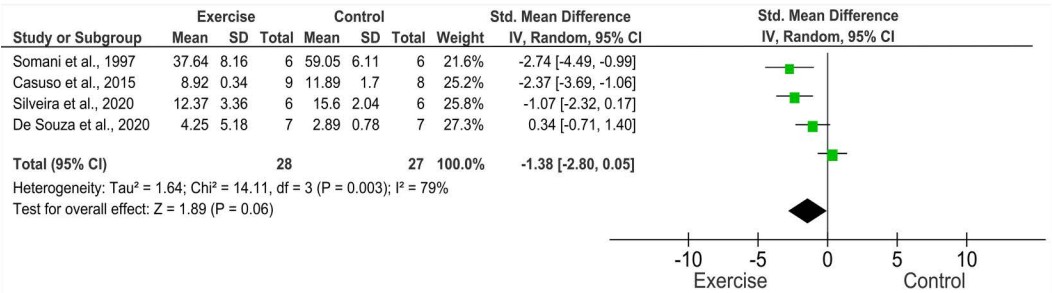

| Study or Subgroup | Exercise Mean | SD | Total | Control Mean | SD | Total | Weight | Std. Mean Difference IV, Random, 95% CI |
|---|---|---|---|---|---|---|---|---|
| Somani et al., 1997 | 37.64 | 8.16 | 6 | 59.05 | 6.11 | 6 | 21.6% | -2.74 [-4.49, -0.99] |
| Casuso et al., 2015 | 8.92 | 0.34 | 9 | 11.89 | 1.7 | 8 | 25.2% | -2.37 [-3.69, -1.06] |
| Silveira et al., 2020 | 12.37 | 3.36 | 6 | 15.6 | 2.04 | 6 | 25.8% | -1.07 [-2.32, 0.17] |
| De Souza et al., 2020 | 4.25 | 5.18 | 7 | 2.89 | 0.78 | 7 | 27.3% | 0.34 [-0.71, 1.40] |
| Total (95% CI) | | | 28 | | | 27 | 100.0% | -1.38 [-2.80, 0.05] |

Heterogeneity: Tau² = 1.64; Chi² = 14.11, df = 3 (P = 0.003); I² = 79%
Test for overall effect: Z = 1.89 (P = 0.06)

# CAT

| Study or Subgroup | Exercise Mean | SD | Total | Control Mean | SD | Total | Weight | Std. Mean Difference IV, Random, 95% CI |
|---|---|---|---|---|---|---|---|---|
| De Souza et al., 2020 | 0.12 | 0.02 | 7 | 0.15 | 0.06 | 7 | 30.6% | -0.63 [-1.71, 0.45] |
| Casuso et al., 2015 | 20 | 0.47 | 9 | 24.01 | 26.39 | 9 | 41.8% | -0.20 [-1.13, 0.72] |
| Somani et al., 1997 | 14.7 | 1.91 | 6 | 14.06 | 1.75 | 6 | 27.5% | 0.32 [-0.82, 1.47] |
| Total (95% CI) | | | 22 | | | 22 | 100.0% | -0.19 [-0.79, 0.41] |

Heterogeneity: Tau² = 0.00; Chi² = 1.40, df = 2 (P = 0.50); I² = 0%
Test for overall effect: Z = 0.62 (P = 0.54)

# GSH

| Study or Subgroup | Exercise Mean | SD | Total | Control Mean | SD | Total | Weight | Std. Mean Difference IV, Random, 95% CI |
|---|---|---|---|---|---|---|---|---|
| Lamarão-vieira et al., 2019 | 99.86 | 8.41 | 10 | 99.02 | 11.2 | 10 | 44.2% | 0.08 [-0.80, 0.96] |
| De Souza et al., 2020 | 85.11 | 14.89 | 7 | 74.41 | 20.93 | 7 | 32.4% | 0.55 [-0.52, 1.63] |
| Somani et al., 1997 | 6.2 | 0.35 | 6 | 5.38 | 0.71 | 6 | 23.4% | 1.35 [0.04, 2.66] |
| Total (95% CI) | | | 23 | | | 23 | 100.0% | 0.53 [-0.15, 1.22] |

Heterogeneity: Tau² = 0.08; Chi² = 2.51, df = 2 (P = 0.28); I² = 20%
Test for overall effect: Z = 1.52 (P = 0.13)

# LPO

| Study or Subgroup | Experimental Mean | SD | Total | Control Mean | SD | Total | Weight | Std. Mean Difference IV, Random, 95% CI |
|---|---|---|---|---|---|---|---|---|
| Chalimoniuk et al., 2015 | 1.77 | 0.72 | 8 | 2.03 | 0.66 | 8 | 51.7% | -0.36 [-1.35, 0.63] |
| De Souza et al., 2020 | 4.5 | 0.8 | 8 | 2.89 | 0.78 | 7 | 48.3% | 1.92 [0.63, 3.21] |
| Total (95% CI) | | | 16 | | | 15 | 100.0% | 0.74 [-1.48, 2.97] |

Heterogeneity: Tau² = 2.24; Chi² = 7.49, df = 1 (P = 0.006); I² = 87%
Test for overall effect: Z = 0.65 (P = 0.51)

**Fig 5. Meta-analysis of moderate-volume PE groups versus controls.**

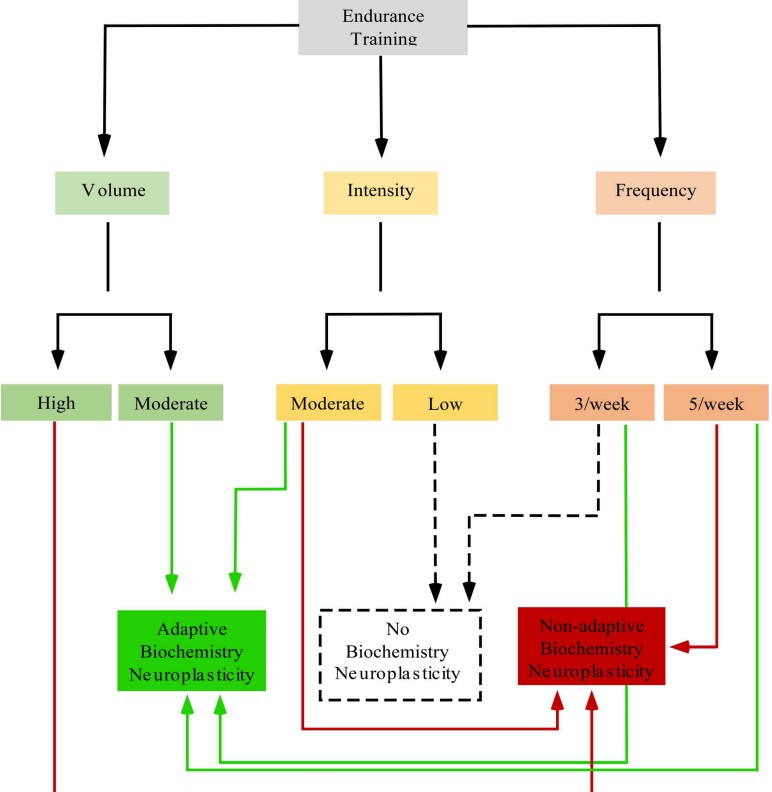

**Fig 6. Schematic drawing of the modulation of PE-induced biochemical neuroplasticity.**

neuroplasticity induced by moderate-intensity PE (Table 1). This meta-analysis revealed that moderate-volume PE did not significantly modulate antioxidant enzyme levels but did significantly reduce LPO [10,14,15,30]. In contrast, LPO increased following high-volume PE [10] (Figs 4 and 5). The high heterogeneity among studies may be attributed to subtle variations in training frequency, volume, or intensity (speed), emphasizing the need for further investigation into the biochemical effects of different PE protocols on cerebellar biochemical neuroplasticity. Conversely, biochemical neuroplasticity was not observed in animals subjected to low-intensity PE protocols [21], see Tables 1 and 2, and Fig 6. Frequency alone did not appear to cause biochemical alterations since protocols with 5 training days per week yielded divergent effects depending on total PE volume [10,14], see Fig 6. It is important to note that studies involving elderly animals and muscle strength training were not included in this review and may yield different cerebellar responses. All selected studies reported aerobic treadmill protocols and presented no high risk of bias in key domains (Fig 2).

PE does not appear to modulate antioxidant enzyme activity in a linear fashion. While some studies reported increased SOD, CAT, GSH, and GR activity [10,15,30], others revealed reduced SOD, CAT, and GSH-Px levels [14,30]. These discrepancies likely reflect transient ROS elevations due to increased energy demands from PE (Table 2). Although excessive ROS levels can damage neurons, moderate levels are essential for neuronal signaling and neuroplasticity [32]. The CNS can tolerate brief ROS increases [33], which activate transcription factors such as Forkhead box O [34,35] that upregulates genes involved in cell survival and differentiation, cell cycle arrest, oxidative stress resistance, and adaptive neuroplasticity [35,36]. This super-compensation mechanism may enhance cerebellar resistance.

Additionally, PE stimulates the expression of neurotrophic factors related to CNS plasticity, such as insulin-like growth factor (IGF), vascular endothelial growth factor (VEGF), nerve growth factor (NGF), and brain-derived neurotrophic factor

(BDNF). These factors reduce oxidative stress by optimizing cellular functions, regulating mitochondrial energy homeostasis, and promoting synaptic regulation [31]. For instance, increased BDNF expression via the PGC-1alpha/FNDC5/irisin pathway [31,37,38] activates the TrkB/BDNF pathway, which activates TrkB tyrosine kinase receptors, and reduces ROS formation [39]. The CREB-BNDF pathway is sensitive to redox reactions and closely linked to the oxidative stress regulatory mechanism since it induces the activation of APE-1 (an inflammation regulator that modulates ROS levels) and thus promotes adaptive biochemical neuroplasticity [2,40,41]. Moreover, BDNF also affects proteasome function, which protects against oxidative stress by degrading damaged and oxidized proteins [42], and thus characterizes adaptive biochemical neuroplasticity (Fig 4).

Neurophysiological responses to PE vary by CNS region [15] and are influenced by intensity, volume, and frequency [17]. Although low-intensity PE does not appear to affect cerebellar biochemical neuroplasticity [21], high-volume PE increases oxidative stress and promotes maladaptive biochemical neuroplasticity [10], see Fig 6. The selected articles reported total running distances to define PE parameters [17]. Volume, intensity, and frequency are adequate variables for accurately assessing PE-induced neuroplasticity [18].

Two studies linked oxidative stress to high-volume PE [10,30]. In one, a single 90-minute PE session increased cerebellar LPO and triggered oxidative stress [10] (see Table 1). Prolonged $O_2$ consumption during PE increases the amount of free radicals and induces phosphorylation of the electron transport chain [43]. The lost electrons in the mitochondria generate superoxide ($O_2^-$), hydrogen peroxide ($H_2O_2$), and hydroxyl (OH) radicals [44,45]. Elevated cerebellar ROS levels impair mitochondrial energetic activity, which modulates redox signaling pathways and causes oxidative imbalance [20]. These events can promote chronic damage in nervous tissue that can lead to cell death and impair cognitive and motor functions [2,14], classifying them as maladaptive biochemical neuroplasticity. Therefore, PE volume plays a pivotal role in determining the direction of cerebellar neuroplasticity.

The threshold at which PE transitions from adaptive to maladaptive neuroplasticity remains unclear and warrants further investigation. This review shows that high-volume PE increases LPO and the risk of maladaptive changes [10], see Figs 4 and 6. However, PE may protect against ethanol-induced oxidative stress in the cerebellum [14].

While oxygen is essential for cellular function, excessive consumption can generate toxic ROS and disrupt biochemical homeostasis [46]. Moderate ROS increases induced by moderate-volume PE are within physiological norms and support adaptive responses [10,46,47]. In contrast, excessive mitochondrial ROS levels (as observed in several diseases) damage lipids, proteins, and DNA, which impairs the maintenance of membrane potential, transmembrane transport, proteostasis, and enzymatic activities [48,49]. These detrimental effects may also occur following prolonged moderate PE.

PE protocols involving less than 80 minutes per day at 5 days per week were effective in reducing LPO and improving some antioxidant activities [10,15,30] (see Table 1). PE activates the PGC-1α pathway, which regulates mitochondrial biogenesis [6], enhances cellular energy metabolism, and increases the cerebellum's resistance to oxidative stress – hallmarks of adaptive biochemical neuroplasticity.

Moderate PE can also attenuate ethanol-induced maladaptive neuroplasticity in the cerebellum (see Table 1). Two studies reported ethanol-induced oxidative stress, which is metabolized to acetaldehyde by the enzyme cytochrome P4502E1 in different CNS regions [50–52]. Damage to DNA, proteins, and particularly highly unsaturated phospholipids modifies the structure and function of neuronal plasma membranes [53]. High cerebellar ROS levels jeopardize $H_2O_2$ decomposition by the P4502E1 enzyme due to iron loss at the catalytic site. In addition, loss of NaDPH cosubstrate impairs the elimination of peroxides formed by ethanol-induced oxidative stress [52], see Fig 5. However, the combination of moderate PE and ethanol consumption promotes a synergistic increase in cytochrome P4502E1 activity as a compensatory response that may be related to a neuroprotective mechanism [14,54]. Excessive ROS levels stimulate the expression of neurotrophic factors while moderate PE regulates energy homeostasis and improves mitochondrial function, which increases the expression of BDNF, FGF, and VEGF [49]. High levels of these factors activate regulatory mechanisms that prevent ethanol-induced oxidative imbalance and thus can be considered adaptive neuroplasticity [55–57], see Fig 7.

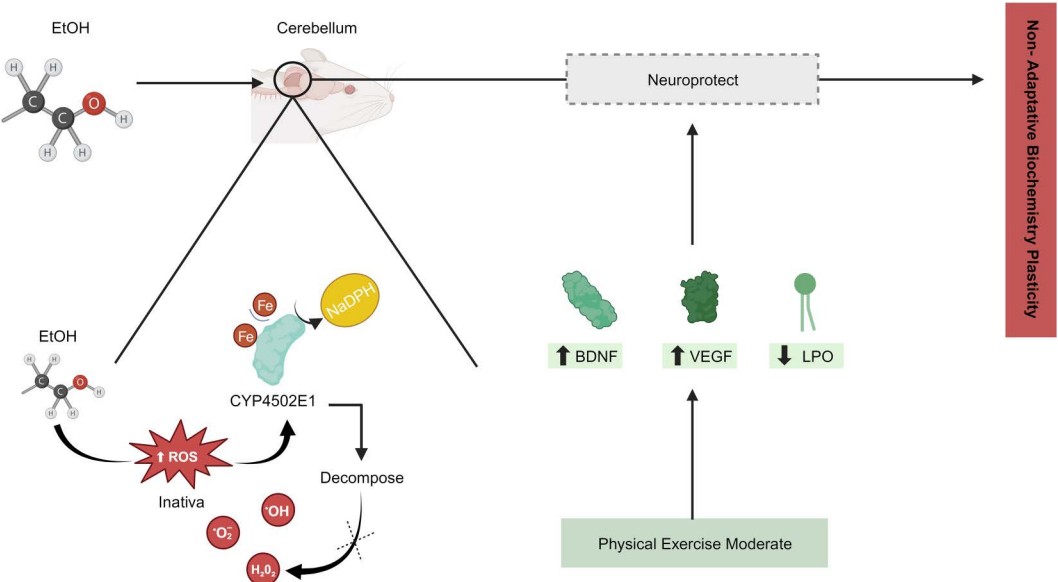

**Fig 7. Schematic drawing of neuroprotective metabolic pathways induced by moderate PE after ethanol consumption.**

Cerebellar biochemical alterations appear to be strongly influenced by PE intensity. No significant alterations were observed following low-intensity PE (~50% of maximal speed) [21], whereas moderate-intensity PE (~70% of maximal speed) significantly reduced cerebellar LPO [14,15], suggesting adaptive neuroplasticity (Fig 6). Although none of the included studies investigated high-intensity PE (Table 1), previous research suggests that CNS can tolerate high $O_2$ consumption only for short periods [15]. According to the American College of Sports Medicine, precise PE variables are essential. Protocols with moderate intensity, duration <80 minutes per day, and frequency of 3–5 days per week appear optimal for inducing adaptive cerebellar neuroplasticity (Table 1). In contrast, high-volume PE (e.g., 90 minutes per day and 5 days per week) induced maladaptive neuroplasticity, representing potential health risks (Figs 6 and 8).

Fine-tuning PE protocols may enhance both the safety and effectiveness of exercise-induced adaptive neuroplasticity. However, this review is limited by the absence of studies evaluating high-intensity protocols. Small sample sizes and between-study heterogeneity may have affected the meta-analytic outcomes, especially for SOD (I² 79%) and LPO (I² 81%) in the moderate-volume group, and SOD (I² 87%), CAT (I² 93%), and LPO (I² 43%) in the high-volume group. While protocol variability (e.g., 4–12 weeks of training) may have contributed to heterogeneity (see Figs 4 and 5), the aerobic-endurance nature of all PE interventions supports the consistency of the neurophysiological responses observed [58].

Human studies using biomarkers and biological fluid components (e.g., SOD, CAT, and GPx) and non-enzymatic anti-oxidants (e.g., GSH and uric acid) have similarly demonstrated that PE intensity, volume, and frequency determine oxidative responses [59–61]. As observed in the animal studies included in this systematic review, low-intensity PE does not significantly alter ROS levels, whereas moderate-to-high-intensity PE performed at moderate volumes reduces oxidative stress and increases antioxidant defenses.

The adaptive/maladaptive neuroplasticity patterns observed in small rodents also appear in humans [59,62]. Moderate PE decreases MDA levels in peripheral blood and increases plasma CAT, SOD, and GPx activity [59,62]. These findings in humans confirm the cerebellar tissue responses observed in small rodents following moderate PE.

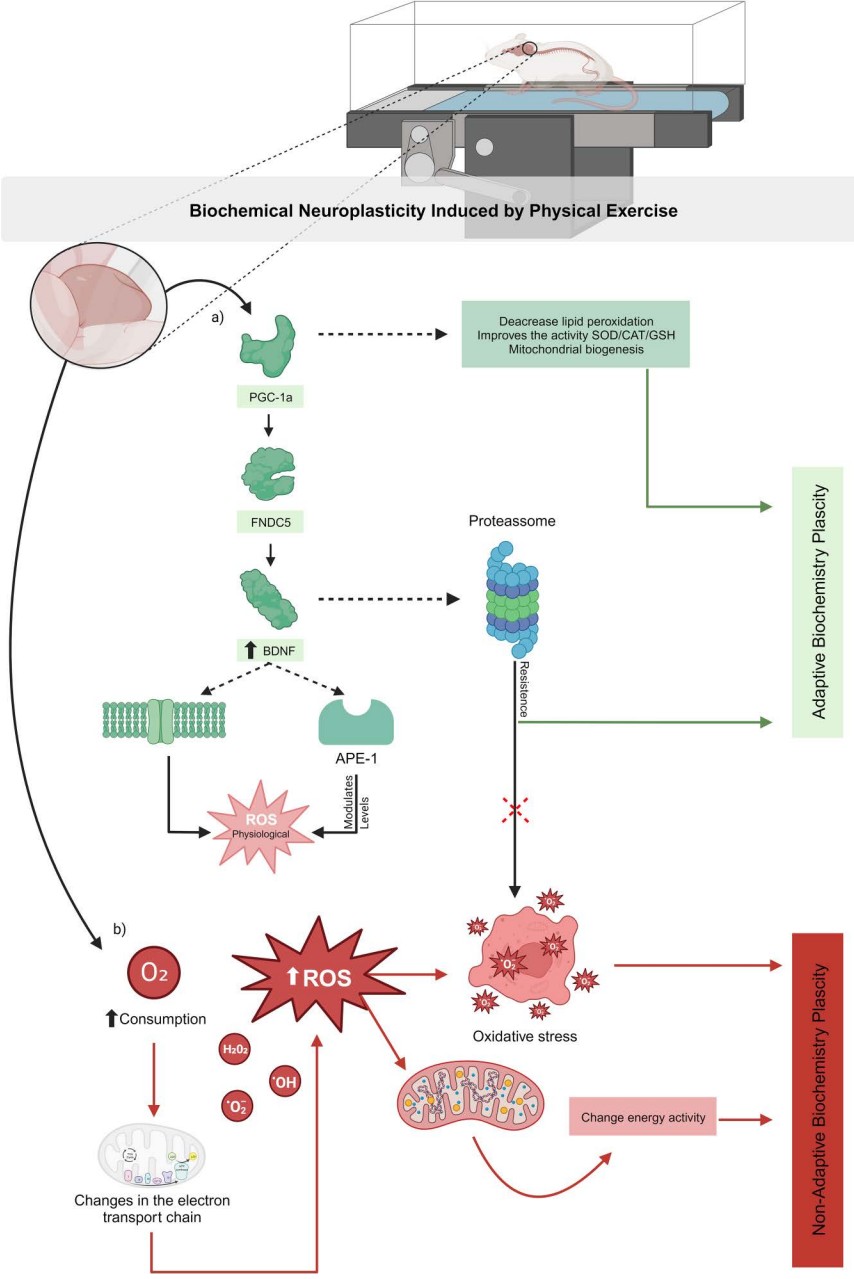

**Fig 8. Schematic drawing of the metabolic pathways involved in PE-induced adaptive (A) and maladaptive (B) neuroplasticity.**

## Conclusion

PE induces either adaptive or maladaptive biochemical neuroplasticity in the cerebellum depending on protocol variables. Low-intensity PE does not induce significant changes, whereas high-volume PE may lead to maladaptive neuroplasticity. In contrast, moderate-volume, moderate-intensity PE performed three to five days per week efficiently promotes adaptive neuroplasticity and protects the cerebellum against oxidative stress. While enzymatic activity responds to cellular changes and limits nervous tissue protection, adaptive biochemical neuroplasticity seems to confer greater resistance and efficiency.

## Supporting information

**S1 File.  PRISMA checklist.**
(DOCX)

**S2 File.  Search Strategy and Study selection.**
(DOCX)

**S3 File.  Study selection.**
(XLSX)

**S4 File.  Data collection.**
(XLSX)

**S5 File.  Risk of bias for individual studies.**
(XLSX)

**S6 File.  Funnel plots.**
(DOCX)

## Acknowledgments

The Deans of Undergraduate and Research/Postgraduate Studies of the Federal University of Pará and the Brazilian Agency for Support and Evaluation of Graduate Education (CAPES) supported this review.

## Author contributions

**Conceptualization:** Marcio Gonçalves Corrêa, Thais Alves Lobão, Carlomagno Pacheco Bahia.

**Data curation:** Gabriel Mesquita da Conceição Bahia, Erica Miranda Sanches Aires, Rebeca da Costa Gomes, Jeffeson Hildo Medeiros de Queiroz, Carlomagno Pacheco Bahia.

**Formal analysis:** Thais Alves Lobão, Gabriel Mesquita da Conceição Bahia, Erica Miranda Sanches Aires, Rebeca da Costa Gomes, Jeffeson Hildo Medeiros de Queiroz, Marta Chagas Monteiro, Carlomagno Pacheco Bahia.

**Funding acquisition:** Marta Chagas Monteiro, Carlomagno Pacheco Bahia.

**Investigation:** Marcio Gonçalves Corrêa, Thais Alves Lobão, Erica Miranda Sanches Aires, Rebeca da Costa Gomes, Jeffeson Hildo Medeiros de Queiroz.

**Methodology:** Marcio Gonçalves Corrêa, Gabriel Mesquita da Conceição Bahia, Erica Miranda Sanches Aires, Jeffeson Hildo Medeiros de Queiroz.

**Supervision:** Carlomagno Pacheco Bahia.

**Visualization:** Carlomagno Pacheco Bahia.

**Writing – original draft:** Marcio Gonçalves Corrêa, Thais Alves Lobão, Gabriel Mesquita da Conceição Bahia, Jeffeson Hildo Medeiros de Queiroz.

**Writing – review & editing:** Marta Chagas Monteiro, Carlomagno Pacheco Bahia.

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
