## [Decision Letter · Decision Letter 0]

Dear Dr. Bahia,

Thank you for submitting your manuscript to PLOS ONE. After careful consideration, we feel that it has merit but does not fully meet PLOS ONE’s publication criteria as it currently stands. Therefore, we invite you to submit a revised version of the manuscript that addresses the points raised during the review process.

We look forward to receiving your revised manuscript.

Kind regards,

Vara Prasad Saka

Academic Editor

PLOS ONE

Journal Requirements: When submitting your revision, we need you to address these additional requirements. 1. Please ensure that your manuscript meets PLOS ONE's style requirements, including those for file naming. The PLOS ONE style templates can be found at https://journals.plos.org/plosone/s/file?id=wjVg/PLOSOne_formatting_sample_main_body.pdf and https://journals.plos.org/plosone/s/file?id=ba62/PLOSOne_formatting_sample_title_authors_affiliations.pdf 2. As required by our policy on Data Availability, please ensure your manuscript or supplementary information includes the following:  A numbered table of all studies identified in the literature search, including those that were excluded from the analyses.   For every excluded study, the table should list the reason(s) for exclusion.   If any of the included studies are unpublished, include a link (URL) to the primary source or detailed information about how the content can be accessed.  A table of all data extracted from the primary research sources for the systematic review and/or meta-analysis. The table must include the following information for each study:  Name of data extractors and date of data extraction  Confirmation that the study was eligible to be included in the review.   All data extracted from each study for the reported systematic review and/or meta-analysis that would be needed to replicate your analyses.  If data or supporting information were obtained from another source (e.g. correspondence with the author of the original research article), please provide the source of data and dates on which the data/information were obtained by your research group.  If applicable for your analysis, a table showing the completed risk of bias and quality/certainty assessments for each study or outcome.  Please ensure this is provided for each domain or parameter assessed. For example, if you used the Cochrane risk-of-bias tool for randomized trials, provide answers to each of the signalling questions for each study. If you used GRADE to assess certainty of evidence, provide judgements about each of the quality of evidence factor. This should be provided for each outcome.   An explanation of how missing data were handled.  This information can be included in the main text, supplementary information, or relevant data repository. Please note that providing these underlying data is a requirement for publication in this journal, and if these data are not provided your manuscript might be rejected. 3. Thank you for stating the following financial disclosure: "CPB: National Council for Scientific and Technological Development – CNPq (grants no. 310054/2018-4, 447835/2014-9, 483404/2013-6, 444967/2020-6, and 444982/2020-5), and the Brazilian Agency for Support and Evaluation of Graduate Education – CAPES (grants PROCAD 21/2018). MCM: National Institutes of Science, Technology and Innovation (PROBIAM Pharmaceuticals Amazonia - INCT/CNPq grant 406819/2022-0), the Brazilian Agency for Support and Evaluation of Graduate Education – CAPES (88882.461690/2019-01), and the Fundação Amazônia Paraense de Amparo à Pesquisa (FAPESPA) grant 005/2016." Please state what role the funders took in the study.  If the funders had no role, please state: ""The funders had no role in study design, data collection and analysis, decision to publish, or preparation of the manuscript."" If this statement is not correct you must amend it as needed. Please include this amended Role of Funder statement in your cover letter; we will change the online submission form on your behalf. 4. Please ensure that you refer to Figure 8 in your text as, if accepted, production will need this reference to link the reader to the figure.

**Additional Editor Comments:**

Clearly state the platform where your protocol is registered.

Use the SYRCLE Risk of Bias tool for animal studies.

Improve the clarity of forest plots and explain how studies from different settings were combined.

Add a quantitative analysis of publication bias (e.g., funnel plot).

Address high heterogeneity and small sample size as limitations in the discussion.

Provide more details on the literature search terms, databases, and inclusion/exclusion criteria.

Discuss calcium’s role in neuroplasticity or explain why it was not included.

Include more detail on the statistical methods used in the meta-analysis.

Expand on how physical exercise affects neuroplasticity mechanisms.

Specify if findings apply only to animal models.

Ensure consistent terminology in the PICO framework and databases.

Add reasons for excluding articles in the PRISMA flowchart.

These revisions will improve clarity and strengthen your manuscript. Thank you.

Reviewers' comments:

Reviewer's Responses to Questions

**Comments to the Author**

1. Is the manuscript technically sound, and do the data support the conclusions?

Reviewer #1: Yes

Reviewer #2: Partly

Reviewer #3: Partly

Reviewer #4: No

Reviewer #5: Yes

2. Has the statistical analysis been performed appropriately and rigorously?

Reviewer #1: Yes

Reviewer #2: Yes

Reviewer #3: No

Reviewer #4: Yes

Reviewer #5: N/A

3. Have the authors made all data underlying the findings in their manuscript fully available?

Reviewer #1: Yes

Reviewer #2: Yes

Reviewer #3: Yes

Reviewer #4: Yes

Reviewer #5: Yes

4. Is the manuscript presented in an intelligible fashion and written in standard English?

Reviewer #1: Yes

Reviewer #2: No

Reviewer #3: No

Reviewer #4: Yes

Reviewer #5: Yes

Reviewer #1: The manuscript titled BIOCHEMICAL NEUROPLASTICITY IN THE CEREBELLUM AFTER PHYSICAL

EXERCISE: SYSTEMATIC REVIEW AND META-ANALYSIS, includes the data related to physical exercise induced biochemical and neurological alterations. I appreciate the authors contribution.

Reviewer #2: There are minor grammatical issues throughout the paper. Careful proofreading is recommended. In my view I suggest that a stronger emphasis on methodological clarity, clearer figures, and more in-depth discussion of the findings would greatly improve the manuscript's impact.

Reviewer #3: Name the open-source platform in which protocol is registered?

Author should assess the risk of bias of animal studies using suitable scale like

The Center for Systematic Review of Laboratory Animal Studies (SYRCLE) Risk of Bias (RoB)

Forest plots are not clear. How studies from different settings were combined?

The quantitative analysis of publication bias should be done.

Quantitative analysis is not convincing as heterogeneity is high as well as number of studies are too less to derive any valid conclusion

Reviewer #4: 1. The research question is incomplete if it is seen in human or animals

2. in PICO, C is mentioned as "Sem physical exercise which is not very clear what it means.

3. There is a mismatch in the names of databases screened mentioned in the manuscript and the search strategy table (lacks Cochrane database/includes Science direct) etc. Kindly make it clear which all databases are screened and how many articles were included from database.

4. There are no proper reasons mentioned in the PRISMA flow chart for exclusion of 1933 articles out of 1943...

Reviewer #5: Please check the typos errors.

Make sure all the references should be uniform.

All abbreviations should be explained for the first time.

Footnotes should be placed below the tables and figures.

Please refer the editors comments.

**Do you want your identity to be public for this peer review?** For information about this choice, including consent withdrawal, please see our Privacy Policy

Reviewer #1: **Yes: ** Phani Kumar Kola

Reviewer #2: **Yes: ** Murali Krishna Moka

Reviewer #3: **Yes: ** Anoop Kumar

Reviewer #4: No

Reviewer #5: No

---

## [Author Response · Author response to Decision Letter 1]

9 Jan 2025

1. The manuscript has been formatted according to the PLOS ONE's style.

2. The following items were added in supplementary information and/or manuscript:

2.1 A numbered table of all studies identified in the literature search, including those that were excluded from the analyses and the reason for exclusion (see Supplementary Material 2)

2.2 There are no unpublished studies included

3. A qualitative and quantitative data extraction table was created for each included study, with the name of the extractors, date of extraction and confirmation of eligibility by a third reviewer (see Supplementary Material 4)

4. There was no search information in other databases, the searches occurred only in databases cited in the manuscript.

5. A completed Cochrane risk of bias table showing the completed risk of bias and quality/certainty assessments for each study or outcome was added (see Supplementary Material 5).

6. There were no missing data, but if there were studies selected with missing data for this systematic review, they would be reported in the data extraction table and taken into account in the risk of bias analysis as recommended in item 8 of the SYRCLE risk of bias table.

7. We added the sentence “The funders had no role in study design, data collection and analysis, decision to publish, or preparation of the manuscript” (see page 24, line 435 of the manuscript)

8. We refer to Figure 8 in the pages 15 (line 248), 16 (lines 271, 273), 18 (line 314), 19 (line 334), 21 (line 378), and 21 (line 386) of the manuscript.

Answers to Additional Editor Comments:

1. We have added the information of the platform where the protocol of this systematic review was registered (see page 4, line 95 - 98 of the manuscript) - 10.17605/OSF.IO/ERBD2.

2. We used the SYRCLE risk of bias tool (see page 6, line 129 - 133 of the manuscript).

3. The quality of figures has been improved and descriptions of the meta-analysis performances have been added in the methods section (see page 7, line 147 - 164 of the manuscript).

4. We quantified the risk of bias using the Higgs statistic (I²) as described in the Quantitative analysis topic (see page 7 of the manuscript). In addition, we improving the quality of the funnel plots (see Supplementary Material 3).

5. We added the limitations of the study in a clearer and more objective way in the discussion topic, such as the high heterogeneity and sample size (See page 22, lines 388 - 395 of the manuscript).

6. Examples of search terms were added to the Eligibility criteria, information, sources and search topic (see page 4, line 100 - 107) of the manuscript and the complete search is available in supplementary material 2. The eligibility criteria and databases used were detailed.

7. The role of calcium ion was not addressed in this systematic review because none of the articles selected and included in this systematic review evaluated or discussed its role in the process of biochemical neuroplasticity, although we recognize the importance of calcium for neuroplasticity mechanisms, especially synaptic plasticity. Although there are still no reports on the role of calcium in biochemical neuroplasticity, this may be addressed in the future.

8. The statistical methods section has been revised and updated, where we better describe the data analysis methods (see the Quantitative Analysis topic, page 7, lines 146 - 164 of the manuscript).

9. Regarding the question raised about how physical exercise affects neuroplasticity mechanisms, this information was better described in detail in the discussion topic.

9.1 – Neuroplasticity events are usually preceded by biochemical changes, which seem to be key mechanisms that can either improve neural function or lead to cell death (5). Changes in enzymatic activities and consequential neurophysiological effects must be understood as biochemical neuroplasticity, the induction of which by PE was demonstrated in this review. Interestingly, variations in the PE protocol can either increase or decrease the activity of antioxidant enzymes and LPO (9). Since several enzymes are involved in biochemistry, PE can promote adaptive or maladaptive biochemical neuroplasticity (Figs. 4, 5 and 8), see page 15 line 242-248.

9.2 – Two studies demonstrated that oxidative stress may be related to high-volume PE practice (9,29). PE for 90 minutes in a single session increases the cerebellar LPO and causes oxidative stress (9), see Table 1. An increase in O2 consumption during prolonged PE can increase the amount of free radicals and provoke several electrochemical reactions such as phosphorylation of the electron transport chain (41). The electrons lost at the transport chain in the mitochondria produce superoxide (O2-), hydrogen peroxide (H2O2), and hydroxyl (OH) radicals (42,43). Increased cerebellar ROS levels alter mitochondrial energetic activity, which is essential for the modulation of redox signaling pathways and causes oxidative imbalance (19). These events can promote chronic changes in the cellular environment of nervous tissue that can lead to cell death and consequently affect cognitive and motor functions (13); thus, they are classified as maladaptive biochemical neuroplasticity. Therefore, the modulation of this PE variable seems to determine the type of cerebellar biochemical neuroplasticity. (see Pages 18 and 19, lines 313 – 325).

9.3 – For instance, increased BDNF expression via the PGC-1alpha/FNDC5/irisin pathway (30,36)) activates mechanisms related to oxidative stress reduction, such as the TrkB/BDNF pathway, which activates TrkB tyrosine kinase receptors and attenuates ROS formation (37). The CREB-BNDF pathway is sensitive to redox reactions and closely linked to the oxidative stress regulatory mechanism since it induces the activation of APE-1 (an inflammation regulator that modulates ROS levels) and thus promotes adaptive biochemical neuroplasticity (38,39). See pages 18 and 19, lines 295 – 302.

10. A paragraph was inserted in the manuscript (page 22, line 399 - 410) that answers the question of whether the results of this systematic review apply only to animal models or can also be applied to humans.

11. We reviewed the supplementary material containing the eligibility criteria and complete search strategy, thus ensuring consistency of the terminology used (See supplementary material 2).

---

## [Decision Letter · Decision Letter 1]

Dear Dr. Bahia,

Thank you for submitting your manuscript to PLOS ONE. After careful consideration, we feel that it has merit but does not fully meet PLOS ONE’s publication criteria as it currently stands. Therefore, we invite you to submit a revised version of the manuscript that addresses the points raised during the review process.

We look forward to receiving your revised manuscript.

Kind regards,

Vara Prasad Saka

Academic Editor

PLOS ONE

**Journal Requirements:**

Reviewers' comments:

Reviewer's Responses to Questions

**Comments to the Author**

Reviewer #2: All comments have been addressed

Reviewer #5: All comments have been addressed

2. Is the manuscript technically sound, and do the data support the conclusions?

Reviewer #2: Yes

Reviewer #5: Yes

3. Has the statistical analysis been performed appropriately and rigorously?

Reviewer #2: Yes

Reviewer #5: Yes

4. Have the authors made all data underlying the findings in their manuscript fully available?

Reviewer #2: Yes

Reviewer #5: Yes

5. Is the manuscript presented in an intelligible fashion and written in standard English?

Reviewer #2: Yes

Reviewer #5: Yes

**Reviewer #2: ** The reviewer comments for the above Manuscript Number: PONE-D-24-33520_R1

Manuscript Title: BIOCHEMICAL NEUROPLASTICITY IN THE CEREBELLUM AFTER PHYSICAL EXERCISE: SYSTEMATIC REVIEW AND META-ANALYSIS

The manuscript presents a study on cerebellar biochemical neuroplasticity, focusing on the role of calcium ions and high-intensity physical exercise. However, it is noted that the study was excluded from studies evaluating these factors, which could be addressed in future iterations.

In study selection you had written “Among 3,107 articles, 767 were duplicates and 2,340 were excluded after the title”. If you see in Figure values are mentioned as 3.107, 2.340 please correct in figure as 3,107 and 2,340.

Can you justify the sentence “2,086 articles were eligible for qualitative and quantitative analyses.” What is the number 2,086 found in abstract part please specify it.

In selection criteria if you would have mention certain time period it will good as my point of view.

In funnel plot diagram looks fine, but please highlight the plot area by using shape fill option in word document, for better view.

In abbreviation section you mentioned as “PE Physical exercise group” in some areas found it as “PE group” clarify and modify accordingly.

The discussion depth on adaptive versus maladaptive neuroplasticity mechanisms is well-emphasized, but more emphasis on real-world applications or translating these findings into human models would strengthen the relevance of the research.

In my view the references would be recent if possible.

The study also discusses the heterogeneity of the included studies, which could be addressed in greater depth.

The study also discusses the high heterogeneity in some metrics, which reduces interpretability and could be improved by discussing the reasons behind this higher variability.

The visual clarity of the figures could be improved, and the supplementary material could be streamlined for key elements.

**Reviewer #5:**  Authors have answered all the queries point wise with proper justification in the revised manuscript raised by reviewers.

**Do you want your identity to be public for this peer review?** For information about this choice, including consent withdrawal, please see our Privacy Policy

Reviewer #2: **Yes: ** Dr Murali Krishna Moka

Reviewer #5: No

---

## [Author Response · Author response to Decision Letter 2]

18 Mar 2025

Dears Reviewers/Editor,

Agradecemos as novas considerações e comentários do revisor para o número do manuscrito seguinte: PONE-D-24-33520_R1. Título do manuscrito: “BIOCHEMICAL NEUROPLASTICITY IN THE CEREBELLUM AFTER PHYSICAL EXERCISE: SYSTEMATIC REVIEW AND META-ANALYSIS”, estamos enviando uma nova versão revisada com as devidas alterações em cada item abordado.

Yours Sincerely,

Carlomagno Pacheco Bahia

---

## [Decision Letter · Decision Letter 2]

Dear Dr. Bahia,

Thank you for submitting your manuscript to PLOS ONE. After careful consideration, we feel that it has merit but does not fully meet PLOS ONE’s publication criteria as it currently stands. Therefore, we invite you to submit a revised version of the manuscript that addresses the points raised during the review process.

We look forward to receiving your revised manuscript.

Kind regards,

Vara Prasad Saka

Academic Editor

PLOS ONE

Reviewers' comments:

Reviewer's Responses to Questions

**Comments to the Author**

Reviewer #2: (No Response)

2. Is the manuscript technically sound, and do the data support the conclusions?

Reviewer #2: Partly

3. Has the statistical analysis been performed appropriately and rigorously?

Reviewer #2: No

4. Have the authors made all data underlying the findings in their manuscript fully available?

Reviewer #2: Yes

5. Is the manuscript presented in an intelligible fashion and written in standard English?

Reviewer #2: No

Reviewer #2: I recommend thoroughly checking the manuscript for the quality

Please check for grammatical and language errors.

Please review the article thoroughly the title in supplementary material 1 it was showing as “BIOCHEMISTRY NEURAL PLASTICITY IN THE CEREBELLUM PROMOTED BY PHYSICAL EXERCISE: A SYSTEMATIC REVIEW AND METANALYSIS” but the main title is BIOCHEMICAL NEUROPLASTICITY can you please justify it.

Can you please define what is “biochemistry changes”

The forest plots are so ambiguous not visible properly

The methodology section needs to be refined

Please make sure the abbreviations are correctly defined PE (I)

Please mention the section uniformly throughout the manuscript “Six out of the 2,340 articles” or “Among 3,107 articles, 767 duplicates and 2,330 articles did not meet the eligibility criteria 172 after title and abstract reading.” Recheck and write in a uniform manner.

Where is the Risk of Bias diagram?

**Do you want your identity to be public for this peer review?** For information about this choice, including consent withdrawal, please see our Privacy Policy

Reviewer #2: **Yes: ** Dr Murali Krishna Moka

---

## [Author Response · Author response to Decision Letter 3]

20 May 2025

Response to reviewers' comments:

Please check for grammatical and language errors.

Authors' response: A grammatical analysis was performed across all text to verify language errors.

Please review the article thoroughly the title in supplementary material 1 it was showing as “BIOCHEMISTRY NEURAL PLASTICITY IN THE CEREBELLUM PROMOTED BY PHYSICAL EXERCISE: A SYSTEMATIC REVIEW AND METANALYSIS” but the main title is BIOCHEMICAL NEUROPLASTICITY can you please justify it.

Authors' response: The title of supplementary material 1 has been corrected to “BIOCHEMICAL NEUROPLASTICITY IN THE CEREBELLUM AFTER PHYSICAL EXERCISE: SYSTEMATIC REVIEW AND META-ANALYSIS”.

Can you please define what is “biochemistry changes”

Authors' response: We have chosen to revise the summary and no longer use the term "biochemistry changes." Note, we have reworded the abstract to the Plos One format.

The forest plots are so ambiguous not visible properly.

Authors' response: We improved the visibility and resolution of the Forest plots, including increasing the size of all graphs. Furthermore, all the images (figures) were updated to the Pace Correct, the Plos One system.

The methodology section needs to be refined

Authors' response: The methods section has been refined according to the journal's standards.

Please make sure the abbreviations are correctly defined PE (I)

Authors' response: The abbreviation was corrected.

Please mention the section uniformly throughout the manuscript “Six out of the 2,340 articles” or “Among 3,107 articles, 767 duplicates and 2,330 articles did not meet the eligibility criteria 172 after title and abstract reading.” Recheck and write in a uniform manner.

Authors' response: The writing has been standardized throughout the entire text.

Where is the Risk of Bias diagram?

Authors' response: The figures have been added as requested.

Note: All changes are yellow highlighted throughout the text.

---

## [Decision Letter · Decision Letter 3]

BIOCHEMICAL NEUROPLASTICITY IN THE CEREBELLUM AFTER PHYSICAL EXERCISE: SYSTEMATIC REVIEW AND META-ANALYSIS

PONE-D-24-33520R3

Dear Dr. Bahia,

We’re pleased to inform you that your manuscript has been judged scientifically suitable for publication and will be formally accepted for publication once it meets all outstanding technical requirements.

Kind regards,

Vara Prasad Saka

Academic Editor

PLOS ONE

Additional Editor Comments (optional):

Reviewers' comments:

Reviewer's Responses to Questions

**Comments to the Author**

Reviewer #2: All comments have been addressed

2. Is the manuscript technically sound, and do the data support the conclusions?

Reviewer #2: Partly

3. Has the statistical analysis been performed appropriately and rigorously?

Reviewer #2: Yes

4. Have the authors made all data underlying the findings in their manuscript fully available?

Reviewer #2: Yes

5. Is the manuscript presented in an intelligible fashion and written in standard English?

Reviewer #2: Yes

Reviewer #2: Minor corrections to be ADDRESSED by the Author The manuscript discusses the topic of how physical exercise (PE) modulates cerebellar biochemical neuroplasticity, offering valuable insights to neuroscience, sports science, and clinical rehabilitation. The systematic review and meta-analysis add rigour to the topic, providing insights valuable to these domains. The article provides an original synthesis of literature, identifying both adaptive and maladaptive responses of the cerebellum to varying PE protocols.

**Do you want your identity to be public for this peer review?** For information about this choice, including consent withdrawal, please see our Privacy Policy

Reviewer #2: **Yes: ** Dr. MURALI KRISHNA MOKA

---

## [Editor Report · Acceptance letter]

PONE-D-24-33520R3

PLOS ONE

Dear Dr. Bahia,

I'm pleased to inform you that your manuscript has been deemed suitable for publication in PLOS ONE. Congratulations! Your manuscript is now being handed over to our production team.

Kind regards,

on behalf of

Dr. Vara Prasad Saka

Academic Editor

PLOS ONE